# AI-Slop to AI-Polish? Aligning Language Models through Edit-Based Writing Rewards and Test-time Computation

**Tuhin Chakrabarty**[1]*  **Philippe Laban**[2]*  **Chien-Sheng Wu**[1]
[1]Salesforce AI Research   [2]Microsoft Research
{tuhin.chakr,wu.jason}@salesforce.com, plaban@microsoft.com

## Abstract

AI-generated text is proliferating across domains, from creative writing and journalism to marketing content and scientific articles. Models can follow user-provided instructions to generate coherent and grammatically correct outputs but in this work, we study a more fundamental question: how do we evaluate and improve the *writing quality* of AI-generated text? Writing quality assessment has received less attention from the community, in part because it is fundamentally subjective and requires expertise. We first introduce the *Writing Quality Benchmark* (WQ) by consolidating five writing-preference datasets into 4,729 writing quality judgments. Our experiments show that most of the competitive baselines, including state-of-the-art LLMs that excel at reasoning tasks, barely outperform random baselines on WQ. We then train specialized Writing Quality Reward Models (WQRM) of various sizes for writing quality assessment that demonstrate strong generalization on four out-of-distribution test sets and 74% accuracy on the WQ benchmark. To further show WQRM's practical benefits during inference, we leverage additional test-time compute to generate and rank multiple candidate revisions, allowing us to select higher-quality outputs from an initial draft. Human evaluation with 9 experienced writers confirm that WQRM-based selection produces writing samples preferred by experts 66% overall, and 72.2% when the reward gap is larger than 1 point. We release our datasets and models to encourage community engagement with writing quality assessment and development of AI writing systems better aligned with human preferences.

## 1  Introduction

Writing is one of the most important pillars of education, enabling learners to critically engage with the topics they study. In *The Rise of Writing* Brandt (2014) argues that the "information economy's insatiable demand for symbol manipulation—'knowledge work'—has forced many workers to reorient their labor around the production of prose" (Laquintano & Vee, 2024). Generative AI tools have further blurred these boundaries, especially around how labor and writing practices are evolving across both academic (Kobak et al., 2024; Lee et al., 2025) and professional contexts (Liang et al., 2025). Often awkward and jarring to read, low-effort text generated by AI is now flooding web browsers and social-media platforms much like spam in old inboxes (Herrman, 2024a; Knibbs, 2024c;d;b;a). This neologistic term of revulsion is often referred to as "A.I. slop" (Herrman, 2024b). Extensive social experimentation with ChatGPT has invited criticism on social media and in the popular news platforms that its writing has a disembodied "robovoice". This has led to humanization methods (Wang et al., 2024) and even start-ups such as StealthGPT or HumanizeAI, which explicitly attempt to make AI-generated text more humanlike.

Despite LLMs showing impressive performance in math and coding, their ability to write high-quality text has been rather pedestrian. Recent work from Chakrabarty et al. (2024b) shows how text generated from widely used LLMs are often rife with clichés, purple prose,

---

**Equal contribution.

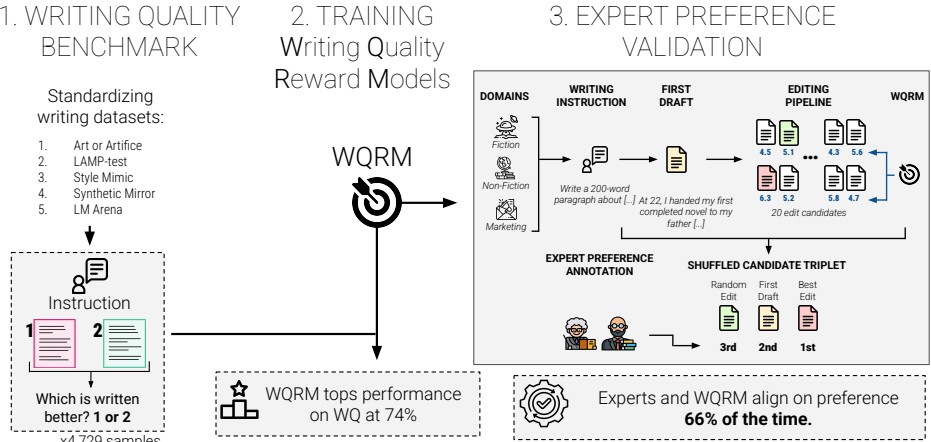

Figure 1: Our three key contributions: (1) A new writing quality benchmark for creative writing evaluation, (2) Writing Quality Reward Models (WQRM) that perform strongly on this benchmark, and (3) Expert validation confirming WQRM aligns with professionals.

poor sentence structure, and unnecessary exposition. This stems from several challenges. Unlike math or coding, writing lacks verifiable rewards. While it would be possible to train a model to write better text by having humans label examples of "good" and "bad" writing, it is challenging due to the required expertise. Self-evaluation using LLMs has proven useful in reward modeling and constitutional AI (Bai et al., 2022), but relying on uncalibrated humans or LLMs for feedback (Lee et al., 2023; Gao et al., 2024) on subjective tasks like writing can lead to reward hacking (Pan et al., 2024) and alignment issues. Recent work from Panickssery et al. (2024) shows the self-aggrandizing nature of LLMs, as evidenced in Table 3 where they prefer their own writing over Nobel Prize winners' work. For the purpose of this paper we define good writing quality as writing that doesn't contain disproportionate amount of peculiar words or phrases, has fewer cliches or hackneyed expressions, is not unnecessarily ornamental as well as doesn't have a overly saccharine and polished tone or voice.

The surge in AI writing assistance demands urgent alignment of AI-generated text with human preferences. Recent work from Gooding et al. (2025) show how LLMs struggle to select high-quality writing actions as judged by human experts, often treating suboptimal and optimal interventions as equally acceptable. They highlight the need for models to better assess the quality and impact of suggested actions, both during generation and across multi-step refinement. Binary preference feedback between paired examples is the most common alignment method for LLMs (Christiano et al., 2017), but it has a significant drawback. The paired outputs may differ in several ways and could be equally worse in terms of quality (Casper et al., 2023; Lambert & Calandra, 2023).[1] Recent work from Chakrabarty et al. (2024b) shows how identifying and editing problematic response segments effectively improves AI alignment. This also reflects the *Reviewing* phase in the cognitive process model of writing (Hayes et al., 1987), where humans evaluate and revise text. They release LAMP (Language model Authored, Manually Polished), a corpus of $1282 < AI - generated, Expert - Edited >$ pairs with implicit preference ($edited > original\_draft$) to improve AI writing (see Table 4 in Appendix A.1). Additionally, each paragraph pair includes normalized scores (1-10) reflecting writing quality before and after editing.

Our work builds on LAMP data to train *Writing Quality Reward Models* (WQRM) across multiple model families using pairwise and scalar rewards. To evaluate WQRM, we introduce the *Writing Quality Benchmark* (WQ), consolidating five datasets that contrast Human-Human, Human-AI, and AI-AI writing pairs reflecting real world applications. In addition to standard reward models we also implement a teacher-student knowledge distillation approach, fine-tuning open-weight models (students) on LAMP with silver rationales generated from

---

[1]Forcing annotators to choose between two undesirable outputs doesn't improve alignment. In the current design of RLHF, annotators are not allowed to pick neither

stronger LLMs (teachers) (Section 3). This framework enhances faithfulness and robustness by transferring reasoning abilities from powerful teachers to efficient students. Empirical results show our LAMP-trained reward models outperform proprietary LLMs like GPT-4o, o1 (OpenAI, 2024), open-weight models like DeepSeek-R1 (Guo et al., 2025), and competitive Reward-Bench models like Skywork-Reward (Liu et al., 2024).

Next, we use expert edit interaction traces from LAMP data (Figure 6) to train a Chain-of-Thought editing model that identifies problematic spans, suggests edits, and combines them into a paragraph with improved writing (Section 5). Following recent work that leverages additional inference-time computation to improve LLM performance (Hosseini et al., 2024; Lightman et al., 2023; Wu et al., 2024; Ji et al., 2025; Snell et al., 2024), we employ *best-of-N-sampling* (Chow et al., 2024; Cobbe et al., 2021; Lightman et al., 2023) to select the best candidate from multiple edited paragraphs based on our reward model. Expert evaluation on LLM-generated responses based on writing instructions across fiction, non-fiction, and marketing confirms the correlation between expert judgment and our reward models. Experts and our best WQRM align in terms of preferences 66% overall, and 72.2% when the reward gap is larger than 1 point. Our results represent progress toward aligning LLMs with expert humans on subjective writing tasks, one of the most common use cases of AI (Handa et al.). As summarized in Figure 1:

- We introduce the *Writing Quality Benchmark* (WQ) by consolidating five writing preference datasets and show how state-of-the-art LLMs and reward models perform close to random chance on writing quality assessment,
- We leverage implicit preference from edits to train competitive open weight reward models (WQRM) of different sizes for judging writing quality. Our reward models achieve top performance on the WQ benchmark,
- We use interaction traces from fine-grained expert edits to train an editing pipeline that improves writing quality. We further leverage additional test-time compute to generate and rank multiple edited paragraphs, allowing us to select higher-quality outputs from an initial draft based on our reward model. Evaluation with professionals confirms that the reward aligns with expert judgments and opens up possible avenues for improving alignment in AI-assisted writing.[2]

## 2 Related Work

**Widespread adoption and Limitations of AI assistance in writing**    Large language models have rapidly transformed written communications across multiple sectors, with approximately 10-24% of text in consumer complaints, corporate communications, job postings, and UN press releases being LLM-assisted by late 2024 (Liang et al., 2025). These adoption rates have stabilized after an initial surge following ChatGPT's release. Outside of technical writing LLMs are also being used for scientific (Liang et al., 2024; Gero et al., 2022) as well as creative writing (Chakrabarty et al., 2024c; Ippolito et al., 2022; Yuan et al., 2022; Mirowski et al., 2023; 2024). Aligning language models with human preferences (Ouyang et al., 2022) has enabled their integration into writing tools such as Google's WorkSpace Labs, Grammarly, and Sudowrite. Despite productivity gains in using AI for writing, several limitations remain with AI-generated text. Prior work (Chakrabarty et al., 2024a;c; Ippolito et al., 2022; Mirowski et al., 2023; Marco et al., 2024) has shown how AI-generated text is often rife with clichés, lacks nuance, subtext, and rhetorical complexity. Through use of syntactic templates Shaib et al. (2024) show the repetitiveness of AI-generated text in comparison to human-written references. More recently Russell et al. (2025) show that AI-generated text is most easily detectable by its characteristic vocabulary, followed by formulaic writing structures and lack of originality. Neither paraphrasing nor humanization effectively removes all of these signatures.

**Human-AI Alignment in Writing**    Recent work from Lee et al. (2024) highlight how LLMs have transformed the processes behind writing, establishing new criteria for future AI writing assistants. Anderson et al. (2024) and Laban et al. (2023) discovered that Large Language Models assisted users in generating more detailed ideas. However, these studies also

---

[2]Our code, data and models are available at `https://github.com/salesforce/creativity_eval/`

found that the outputs were less semantically distinct across different users (Padmakumar & He, 2023), and participants reported feeling diminished responsibility for the ideas they produced.In a similar vein Li et al. (2024) explores people's attitudes toward AI writing assistants, finding that while many value and prefer AI assistance for creative tasks and productivity gains, this comes with potential drawbacks in reduced accountability and diversity in writing outcomes. Liu et al. (2025) introduce eRevise+RF, an automated writing evaluation system designed to assess student essay revisions and offer formative feedback. The system was deployed with 406 students across three schools, demonstrating effectiveness in evaluating evidence usage, identifying revisions, and determining revision success. Prior work from Pan et al. (2024) shows language models can enhance outputs through feedback. However, iterative self-refinement using another language model as evaluator may lead to reward hacking, where models exploit evaluator weaknesses. Chakrabarty et al. (2024b) shows how LLMs across different model families share common writing idiosyncrasies and how automatically editing these idiosyncrasies improves alignment, based on a behavioral study with 12 writers.

Unlike prior work that has focused either on detecting/addressing issues in AI writing our work introduces Writing Quality Reward Models (WQRMs) trained on expert edits that outperform state-of-the-art LLMs on a Writing Quality benchmark.

# 3 Writing Quality Reward Models

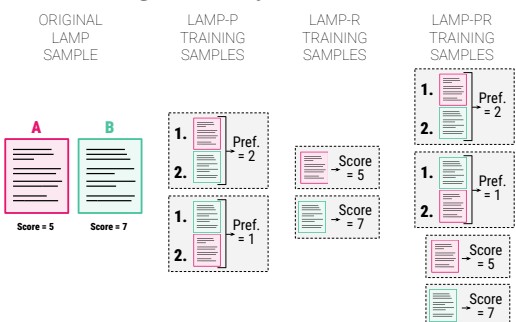

Figure 2: Transforming LAMP annotations into classification and regression data points used during fine-tuning of WQRM models.

We rely on the LAMP (Language model Authored, Manually Polished) corpus from Chakrabarty et al. (2024b) to train reward models. As illustrated in Figure 2, each sample in LAMP consists of a writing instruction and two paragraphs that match this instruction. The paragraphs in LAMP range from 150 to 400 words, and span across fiction and non-fiction. Table 4 in Appendix A.1 shows a sample from LAMP, highlighting the edits implemented by an expert to improve writing quality. We use three methods to transform LAMP samples into training and validation data points for our models: pairwise (**P**), scalar (**R**), and combined (**PR**). With the P method, each data point presents two paragraphs as input (1 and 2) and requires a binary classification output indicating which paragraph has higher writing quality (i.e., the output is 1 or 2). Each LAMP sample is duplicated into two P data points by considering both paragraph orders (AI-generated, Expert-Edited → 2) and (Expert-Edited, AI-generated → 1). With the R method, each data point takes a single paragraph as input and outputs a regression value predicting the quality score of that paragraph. Since each LAMP sample contains two paragraphs (before and after edit), it generates two R data points. The PR method combines both approaches, yielding four data points per LAMP sample (two from P and two from R). There are a total of 1,282 samples in LAMP, and we follow the author's split divisions of 1,000 training, 67 validation, and 215 test samples. Applying the data transformation described above, the P, R, and PR variants of the training data we obtain consist of 2,000, 2,000, and 4,000 training data points, respectively. For our experiments, we trained both generative LLMs (Llama3.1 (Dubey et al., 2024)) and encoder-only models (ModernBert (Warner et al., 2024)).

**Encoder-Only WQRM**   We follow the standard approach introduced in the original BERT paper (Devlin et al., 2019) to add and finetune two task-specific heads to a ModernBERT-Large model (Warner et al., 2024). The input data points contain either one paragraph (for R data points) or two paragraphs (for P data points), which are encoded jointly with a pre-defined separator token when needed. For each paragraph, we compute a "paragraph vector" by pooling the last layer's activations across all tokens in that paragraph. These paragraph vectors serve as input to either a regression (R) or classification (P) head. The

regression head transforms the vector through a learned linear projection from the model's inner dimension to a scalar, followed by a scaled sigmoid to align with the 1-10 score range. The classification head is aparametric, using a cosine similarity operation between the two paragraph vectors. We use mean-squared error loss for R data points and cross entropy for P data points. Following convention for encoder-only models, we finetune the entire model's weights (Devlin et al., 2019). We selected ModernBERT-Large, the largest available model, for our experiments. We fine-tuned three variants: **MBERT-WQRM-P**, **MBERT-WQRM-R**, and **MBERT-WQRM-PR**, each on their corresponding data variants. Hyperparameters, including learning rate and number of epochs, were optimized by minimizing validation loss. PR models can be used in either P- or R-mode at test-time. Initial evaluation indicated that PR models achieve higher performance in R-mode, and as such we used all PR models in R-mode by default during evaluation.

**Generative WQRM** We finetune generative transformer architectures by converting classification and regression tasks to sequence-to-sequence problems using JSON output format (Table 5). We employ QLora (Dettmers et al., 2023) parameter-efficient tuning with FSDP (Zhao et al., 2023) and cross-entropy loss. Generative methods can produce natural-language rationales alongside predictions for interpretability. Wiegreffe et al. (2020) demonstrated label-rationale association as essential for response faithfulness, while (Ludan et al., 2023; Hase & Bansal, 2021) argued for incorporating explanations in model input/output to improve robustness against spurious cues. Since LAMP lacks expert rationales, we augment it with LLM-generated silver rationales. We collected five examples from professional writers showing either paragraph strength contrasts (P-style) or holistic critiques/praise (R-style), instructing them to cite specific excerpts. These expert rationales serve as demonstrations for Claude3.5 Sonnet[3] to generate rationales (examples in Table 6, Appendix A.3).

The rationale augmentation is then used in two variants, either providing the rationales on the input (IR→O), or requiring the generative model to produce the rationale as part of its output (I→RO). We note that rationales are not available at test-time, and are only included during training as an augmentation technique. We finetune a total of seven variants, all based on LLama 3.1 70b model: Llama-WQRM-P, Llama-WQRM-R, Llama-WQRM-PR, Llama-WQRM-P-IR→O and Llama-WQRM-P-I→RO, Llama-WQRM-PR-IR→O and Llama-WQRM-PR-I→RO, based on different versions of the training data, and tune hyperparameters by minimizing validation loss.

## 4 The Writing Quality Benchmark

| Dataset | Pair Origin | Annotator | Len | N |
|---|---|---|---|---|
| Art or Artifice | 🖥🖥/🖥🧍 | Expert | 1.5-3k | 144 |
| LAMP-test | 🖥🖥/🖥🧍 | Expert | 200-400 | 1,206 |
| Style Mimic | 🧍🧍 | Expert | 200-400 | 300 |
| Synth. Mirror | 🖥🧍 | Expert | 200-400 | 1,120 |
| LM Arena | 🖥🖥 | Crowd | 200-2.5k | 1,959 |

Table 1: Writing Quality benchmark composition. Pair Origin: evaluated pairs are AI-generated (🖥) or human-written (🧍); Len: #words in evaluated responses; N: total evaluation pairs contributed to the benchmark.

We create the first benchmark centered on the task of writing quality assessment by collecting five relevant datasets and standardizing their data formats into a pairwise preference task. The task in the benchmark consists of a writing instruction and two writing responses, with a binary label indicating which of the two responses has higher writing quality. Table 1 lists the five datasets we selected for the benchmark, along with key properties of each dataset that lead to a comprehensive benchmark for writing quality. We include three datasets that involve AI-AI comparisons (Art or Artifice (Chakrabarty et al., 2024a), LAMP-test (Chakrabarty et al., 2024b), and LM Arena (Zheng et al., 2023)), three that involve AI-Human comparisons (Art or Artifice, LAMP-test, and Synthetic Mirror), and one that involves Human-Human comparisons (Style Mimic) (Anonymous, 2025). This diversity ensures that models that perform well on the benchmark can judge writing quality regardless of whether the response was LLM generated or human-written.

To assess writing quality prior work has argued for evaluation by professionals (ones with writing experience). Nevertheless, some writing quality preference datasets are based on

---

[3]Considered a top-performing model for writing tasks at the time of experiments.

crowd-sourced judgments. We include four datasets based on expert judgments and one dataset based on crowd-sourced annotation (LM Arena) to represent both perspectives in the benchmark. Finally, we selected two datasets with long responses (Art or Artifice, LM Arena) and three with shorter responses ranging from 200-400 words, ensuring that models that perform well on the benchmark are capable of judging writing quality irrespective of length. Appendix A.4 details the procedure we followed to extract and standardize each dataset. Appendix A.5 provides an analysis we conducted on the relative difficulty of each dataset in the benchmark, finding that the five selected datasets provide a breadth of coverage in terms of difficulty.

| | **Writing Quality Benchmark** | | | | | |
|---|---|---|---|---|---|---|
| **Model** | Synthetic Mirror 🖐👤 | Art or Artifice 🖐🖐/🖐👤 | LAMP 🖐🖐/🖐👤 | Style Mimic 👤👤 | LM Arena 🖐🖐 | **Overall** (↑) All |
| 🌀 MBERT-WQRM-PR | 99.8 | 80.6 | 72.6 | 67.3 | 51.0 | 74.3 |
| 🌀 MBERT-WQRM-R | 100.0 | 80.6 | 76.1 | 59.3 | 51.0 | 73.4 |
| 🌀 MBERT-WQRM-P | 99.5 | 54.2 | 71.2 | 67.0 | 46.8 | 67.7 |
| 🌀 Llama3.1 - P — IR → O | 100.0 | 80.5 | 74.9 | 43.0 | 52.8 | 70.2 |
| 🌀 Llama3.1 - PR — IR → O | 99.6 | 69.4 | 73.7 | 54.3 | 50.1 | 69.4 |
| 🌀 Llama3.1 - PR — I → OR | 99.1 | 76.3 | 71.7 | 42.6 | 55.2 | 68.9 |
| 🌀 Llama3.1 - P — I → OR | 99.9 | 75.1 | 74.1 | 38.6 | 49.1 | 67.3 |
| 🌀 Llama3.1 (70b) - PR | 94.8 | 52.0 | 71.3 | 40.6 | 44.3 | 60.6 |
| 🌀 Llama3.1 (70b) - P | 88.1 | 45.1 | 71.7 | 35.6 | 47.7 | 57.6 |
| 🌀 Llama3.1 (70b) - R | 44.8 | 50.0 | 40.3 | 50.0 | 54.3 | 47.9 |
| 🔍 Pangram | 100.0 | 72.6 | 56.5 | 47.3 | 48.4 | 65.0 |
| ✒ O3 | 67.7 | 85.4 | 41.4 | 67.5 | 59.6 | 64.3 |
| 🎚 Skywork-8B-v0.2 | 90.3 | 68.1 | 54.2 | 34.0 | 55.8 | 60.5 |
| ◎ GPT-4o (5FS) | 39.5 | 68.8 | 40.3 | 67.3 | 55.5 | 54.3 |
| ✒ O1 | 25.8 | 67.4 | 39.8 | 68.7 | 56.7 | 51.7 |
| ✒ DeepSeek-r1 | 31.5 | 54.9 | 39.2 | 47.3 | 57.0 | 46.0 |
| ✒ GPT-4o | 7.5 | 56.2 | 37.8 | 47.7 | 55.4 | 40.9 |

Table 2: Writing Quality Benchmark results. We evaluate ✒ zero-shot and ◎ fewshot LLMs, 🎚 generic reward models, 🔍 AI-detection models, and our 🌀 fine-tuned models.

## 4.1 Experimental Results on WQ

Our experiments on the WQ benchmark include four classes of models. First, Zero-Shot (ZS) and Few-Shot (FS) methods with top-performing instruction-tuned LLMs. We included both non-reasoning (GPT-4o) and reasoning models (Deepseek-R1, O1). Second, a top-performing generic reward model – SkyWork-8b-v0.2 – based on results on the RewardBench leaderboard (Lambert et al., 2024). Third, we include the Pangram AI-detector [4], accessed through API. Finally, the trained WQRM models in generative and encoder-only settings as described in Section 3. Models that can produce pairwise judgments (such as SkyWork or WQRM-P models) were used as is, but for models that produce scalar rewards (WQRM-R, Pangram), a scalar reward was computed for each response, and inequality was applied to emit a pairwise preference. Scalar rewards can theoretically lead to a tie (a score difference of less than an epsilon like 0.001), but we observe few of these in practice (less than 0.1% of pairs), and resolve those randomly.

Experimental results are summarized in Table 2. First, we find that all the LLMs used in zero-shot settings perform below or a few percentage points above a random baseline of 50%. The performance is particularly low on portions of WQ that involve AI-human preference pairs. This confirms prior findings that **LLMs used in LLM-as-a-judge settings tend to prefer AI-generation over human-writing** (Panickssery et al., 2024). The O1 and R1 reasoning models do not significantly outperform their non-reasoning counterparts, indicating that out-of-the box COT-style reasoning, useful for math or coding tasks doesn't improve writing quality assessement. O3 shows improvement on Synthetic Mirror and Art or Artifice showing some promise.Finally, adding five few-shot examples to GPT-4o does help improve performance from 40.9 to 54.3, however further experiments with additional

---

[4] https://www.pangram.com/dashboard?type=text

in-context examples did not lead to further gains, confirming that **few-shot examples in the instruction are not sufficient to achieve strong performance on WQ**.

The generic reward model – Skywork-8b-v0.2 – achieves an overall accuracy of 60.5, with strong performance on Synthetic Mirror and Art or Artifice. Though better than random, the overall performance is much lower than the 93% performance the model achieves on RewardBench, indicating that **reward models geared for instruction-following evaluation are not effective at writing quality assessment out-of-the-box**.

The Pangram AI detection system achieves a total performance of 65.0%, the top performance for untrained models. Pangram achieves near-perfect performance on Synthetic Mirror and the AI-Human pairs of Art or Artifice. On samples that do not involve distinguishing between AI and human text, Pangram achieves near-random performance. In other words, **AI-detection tools only correlate with writing quality assessment when an AI-generated text is judged to be worse than human-written text**.

Finally, the trained WQRM models achieve top-performance on the benchmark. The Llama-based models achieve their strongest performance in the IR→O settings, confirming that augmenting the training data with rationales is beneficial, with models that can generate rationales alongside their prediction. **The ModernBERT-based models achieve the highest overall accuracy of 74.3%**, with the PR variant outperforming the P and R models, indicating that pairwise and reward-based training can be complementary. While its surprising to see a smaller model outperform Llama3.1-70B it could be due to PEFT or the way the loss function is optimized. Future work can focus on bridging this gap.

We observe that generative WQRM models perform best in P-mode, whereas encoder models perform best in R-mode. We emit a hypothesis for this reversal of relationship, related to the choice of loss. The generative models (Llama) are trained with a sequence-to-sequence loss, whereas the encoder-only models (MBert) are trained with custom losses (pairwise classification for P, mean-squared error for R). In other words, LLama training on the reward-based data is more similar to 10-way classification than actual score regression, whereas the MBert training makes better use of the reward-based data. This leads the MBERT-R models to outperform MBert-P models, whereas the reverse is true for the LLama models, as they are not able to properly take advantage of the R-based data.

Looking at performance on individual datasets, Synthetic Mirror is the the easiest dataset, with eight models achieving near-perfect performance. Some models achieve 80%+ performance on Art or Artifice, indicating that long-context evaluation is challenging but achievable. Style Mimic and LM Arena are the most challenging in terms of accuracy. Style Mimic is likely challenging as it is the only dataset that involves comparisons that do not involve AI-generated text, but two relatively high-quality human-written candidates. LM Arena is challenging to all systems, with top performance at 57% by Deepseek-R1. This low performance could be due to the crowd-sourced nature of LM Arena, with the dataset representing much broader and potentially noisier judgments. Though our trained WQRM models outperform baselines by almost 10%+ overall, there remains wide room for improvement: **writing quality assessment remains an open challenge to the community**. Additional analysis in upcoming Sections refers to the top-performing model – MBERT-WQRM-PR – simply as WQRM.

## 5  Editing Pipeline with Test-Time Compute

To better understand the practical value of the WQRM model, we integrate it into a text-editing pipeline to produce LLM-generated candidates of higher-quality according to WQRM scores. We first introduce the editing pipeline and candidate generation procedure, and then describe the large-scale preference annotation we conducted with professional writers to validate WQRM as part of an editing pipeline.

### 5.1  Generating edits via Supervised Finetuning

Prior work from Chakrabarty et al. (2024b) shows experimentally that LLMs' text idiosyncrasies (cliches, redundancy, lack of subtext, etc.) can be mitigated through self-editing in an in-context setup. Borrowing motivation from them we teach LLMs how to improve

their response via edits. Figure 6 illustrates the three components of the editing pipeline. Given a first draft response to an instruction from any given LLM, the first step consists of identifying and listing *idiosyncrasies*: spans in the first draft that can be rephrased to improve overall writing quality. For each identified idiosyncrasy, a second stage consists in rewriting the idiosyncrasy. This is framed as an *executable edit* (Laban et al., 2023), where each edit consists of replacing an original string in a draft with an improved version. The third step simply executes all edits (by applying a series of string replace operations) to obtain the final edited draft. While Chakrabarty et al. (2024b) implemented this through prompt-chaining (Wu et al., 2022) with few-shot examples, we improved efficiency by supervised fine-tuning of GPT-4o and Llama3.1 70B based on the entire LAMP training set. The training input consists of the first draft alongside the entire edit interaction trace (detect, rewrite, execute) in a step-by-step chain of thought prompt, and the output is the edited paragraph. See Appendix A.7 for an example COT prompt.

### 5.2 Selecting edited response by leveraging Test-Time Compute

Recent work from Snell et al. (2024) shows that test-time compute can be scaled optimally by using a reward model to search over the space of solutions. This approach typically involves generating multiple candidate responses and using a verifier to select an optimal response (Cobbe et al., 2021). The most popular technique to increase test-time compute is Best-of-N sampling also known as Rejection Sampling, in which N candidates are generated independently. The reward model is then used to score each candidate, and the top-scoring candidate is selected. While test-time scaling is effective for reasoning tasks, our work aims to measure whether it is a practical strategy to improve human-AI alignment in subjective tasks such as writing.Next we describe the validation study with experts to measure how well calibrated our WQRMs are to human judgment and whether additional test-time computation leads to meaningful improvements in AI writing quality.

## 6  How well calibrated are our reward models ?

We generated 100 draft responses (50 GPT4-o, 50 Llama3.1 70B) based on 90 writing instructions spanning 3 domains: literary fiction, non-fiction, and product marketing. For literary fiction and non-fiction we create the instructions through *instruction back-translation* (Li et al., 2023) conditioned on expert-written paragraphs in Anonymous (2025) and news articles in the data from Russell et al. (2025). Marketing writing instructions were based on products recommended in WireCutter articles[5] across the Home, Kitchen and Tech sections. The right portion of Figure 1 summarizes the process we follow to leverage test-time compute. Specifically, we obtain a first draft from a LLM (GPT4o or Llama3.1 70B) followed by drawing $N = 20$ candidate edited responses from the respective SFT model (Section 5.1) [6], and score each candidate with the WQRM model. We filter out any candidate that scores lower than the first drafts, and then form response triplets by selecting the *first draft*, a randomly-selected edited response (*random edit*), and the Best-of-N candidate response according to WQRM (*Best Edit*) (See example triplet in Table 9). We recruited 9 professional writers through mailing lists from top MFA programs in the US. They were asked to rank three responses based on its overall quality (See Figure 8 for interface). Each response triplet were annotated by three experts, which we aggregated into a majority rank. Participants completed annotation in batches of 10 triplets at a time, and were paid $100 per batch.

### 6.1 Study Findings

Figure 3 summarizes findings from the expert annotation. In Figure 3a, we plot the distribution of rankings across all triplets. Best Edit candidates were most preferred overall with an average rank of 1.58, followed by random edit (2.09) and first draft (2.26). The breakdown of rankings across domains (fiction, non-fiction, marketing) or LLM (GPT-4o vs. Llama 3.1) is presented in Appendix A.8. In short, **Best Edit achieves the top rank in all conditions, confirming the generalization of WQRM scores across conditions.**

If the reward model is well-calibrated, the WQRM score gap between responses should indicate their qualitative difference. For example, responses scoring 4 and 6 should have a larger

---

[5]https://www.nytimes.com/wirecutter/

[6]If first draft is from GPT4o we use GPT4o SFT model

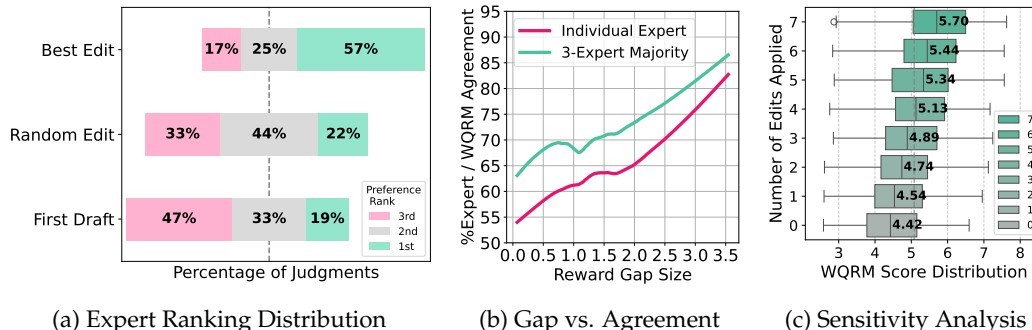

(a) Expert Ranking Distribution     (b) Gap vs. Agreement     (c) Sensitivity Analysis

Figure 3: Results and analysis of WQRM based: (a) distribution of preference based on 300 expert triplet rankings, (b) calibration between gap in WQRM scores and matching expert preference, and (c) applying experts edits gradually to a draft leads to gradual reward gains.

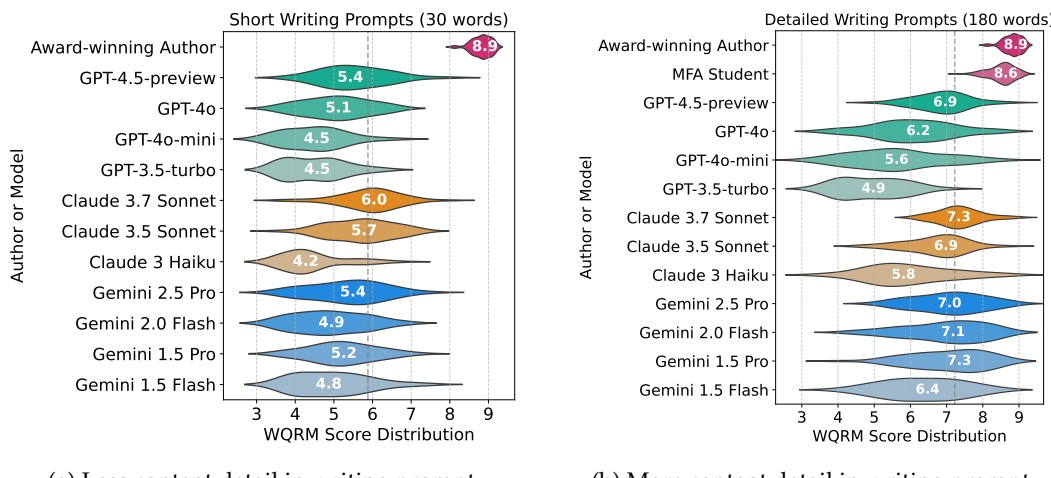

(a) Less content detail in writing prompt     (b) More content detail in writing prompt

Figure 4: Writing quality analysis of human-written and LLM-generated texts according to WQRM on (a) less and (b) more content detail in the writing prompt. Prompts with less content detail average 30 words, whereas prompts with more content detail average 180.

quality gap than those scoring 4 and 4.5. To inspect WQRM calibration, we computed the WQRM gap between all annotated response pairs and plotted it against expert annotation agreement. As shown in Figure 3b, WQRM gap positively correlates with expert agreement: when responses differ by $\leq 0.5$ points, individual experts prefer the higher-scoring response only 55% of the time. When the gap exceeds 3.0, this increases to 80%. Agreement with majority rank based on three expert annotations (green line) shows even stronger positive correlation. **In short, we find evidence that WQRM is well-calibrated: a wider gap in scores between two responses is evidence that an expert (or group of experts) would be more likely to prefer the higher-scoring response over the lower-scoring response.**

Besides calibration, we analyze the sensitivity of the WQRM model to minor edits and their impact on writing quality. The LAMP dataset consists of drafts that are edited by expert writers to improve writing, with samples comprising of eight edits per passage on average. We implement a *gradual* version of the LAMP-test set, where each expert edit is reversed, and we execute them one at a time, computing the WQRM score at each intermediate step. Results from the gradual LAMP-test are summarized in Figure 3c: each time an additional edit is implemented, the median WQRM score increases by 0.2, even though WQRM was not trained on intermediate responses and only saw samples where no edit or all edits have been applied. **In summary, we find evidence that minor edits to a response will lead to small but significant changes in WQRM scores, indicative of a fine sensitivity of the reward model.**

# 7 How does content affect writing quality?

Effectively judging writing quality impacts both understanding and improving LLM writing. Writing quality is however closely tied to content. Its known that LLMs struggle with novel ideas (content planning), making their writing appear trite. Even with detailed original content, they struggle to maintain good writing standards (avoiding clichés, revealing subtext, and introducing purple prose). To understand how content affects writing quality, we analyzed writing from several LLMs with and without detailed content. We used 50 writing instructions from Style Mimic data, creating two variants: a 30-word prompt with less detail (e.g., "A family Christmas unfolds through emotional reflections on a father's new family, a daughter's excuse to stay behind, and the complex dynamics of grief and blended identities.") and a 150-200 word detailed prompt (Table 10 in Appendix). Style Mimic provides an original excerpt from an award-winning author and an MFA student's attempt to mimic that style for each prompt. Each sample includes the detailed content used for 4b.

Since WQRM was only trained on samples from LAMP, which consists of AI-generated paragraphs edited by MFA students, we retrained a better calibrated reward model with few fully human written high quality text (See Appendix A.11 for more details).Figure 4a shows writing quality scores from the WQRM model when prompts lack detailed content. Award-winning authors achieve a median score of 8.9, while LLMs score 4.8-6.6 with much higher variance. Despite WQRM being trained only on AI-generated paragraphs edited by MFA students and relatively fewer human written samples, it scored 50 author-written texts higher than all LLMs, demonstrating model generalization. GPT4.5, though considered the best writing LLM, showed no quality advantage.The significant gap between award-winning authors and LLMs shows that **in the absence of original good-quality content, all LLMs are poor writers.**

Figure 4b shows the writing quality of several LLMs leveraging the new WQRM model when detailed content is provided in the writing prompt. As a matter of fact the content detail is often 0.5x to 0.75x times the word count of the paragraph to be written/generated. Results with the detailed prompts provide additional insights. Though the variance remains high for all models, the more recent models (GPT-4.5, Claude 3.7-Sonnet, Gemini-2.5-pro) achieve improved writing quality given the more detailed prompts, achieving median scores of around 7.0. This should not be surprising as the amount of details provided in the writing prompt reduces the burden for originality and novelty from the LLM. What is particularly impressive here is paragraphs written by MFA students based on the same detailed content were rated significantly higher than all LLMs with a median of 8.6. The gap between award-winning authors and MFA students is narrow here, although the distribution from MFA students shows higher variance. Our results highlight that **even when provided with very detailed original content, LLMs are far behind trained writers**.

In summary, the analysis reveals that current LLMs are not yet capable of reliably generating high-quality creative writing at the level of an MFA student or award-winning author, especially when not spoonfed with original content. When provided with enough content detail in the prompt, the latest models show promise but still remain unreliable.

# 8 Conclusion

In this work, we introduced the Writing Quality benchmark (WQ) and Writing Quality Reward Models (WQRM) to address the critical challenge of evaluating and improving the quality of AI-generated text. Our models trained on implicit preference via edits significantly outperform existing approaches, achieving 74% accuracy on the WQ benchmark and demonstrating strong generalization across diverse writing contexts, as confirmed by a validation study involving 9 professional writers. Future work can address alternative test time computation such as long chains-of-thought (CoTs) enabling strategies like backtracking and correction of idiosyncrasies for improving writing. While our approach improves AI generated text by reducing idiosyncrasies, it is no where near expert quality writing. However, we hope that our contributions can serve as a catalyst for further research in writing quality assessment and the development of AI writing systems that are more aligned with human preferences.

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

# A  Appendix

| Which is better quality writing? | |
|---|---|
| **A**: One evening – it was our last day – in Tours, we had dinner in a brightly-lit restaurant where the walls were lined with mirrors, frequented by a sophisticated clientele. [.......] made with "pig slop," white and tasteless. [.......] expressing resentment at having been treated with contempt because we were not chic customers who ate "à la carte." | Paragraph B is of higher quality

**Conciseness and Clarity**: Paragraph B is more streamlined[...] tighter and more focused.

**Emotional Depth**: Paragraph B captures the speaker's evolving realization [....] making the writing more impactful. |
| **B**: We sat at the group table, my father and I, our reflections dim in the restaurant's mirror. The waitress barely glanced our way. I adjusted my glasses, [....] I realize now that what I witnessed that day was not just a meal but the precise moment when social class became visible to me, manifested in a pot of yogurt, in the waitress's averted eyes, [....]. | **Imagery and Language**: The use of phrases [....] adds nuance and emotional weight. In contrast, Paragraph A's descriptions is literal and lack [...]

**Pacing and Flow**: The flow of Paragraph B is smoother [...] Paragraph A, [....] frag mented with less of a clear emotional journey.

Overall, Paragraph B delivers a more poignant [...] |

Table 3: LLM preferring its own writing (Paragraph B) over Paragraph A written by Annie Ernaux, 2022 Nobel Prize in Literature.

## A.1  Example LAMP Sample

Table 4 shows an example from LAMP data with writing prompt and response before and after edit.

| | |
|---|---|
| Prompt | What happened while you were sitting in your car, waiting for a call, amidst the torrential rain and chaotic scene in La Conchita? |
| Edited Resp. | As I sat in my car, the rain drummed against the roof ~~, a relentless beat that mirrored the anxiety thrumming through my veins~~. I was waiting for a call, any call, that would give me a reason to move, to act ~~, to escape the suffocating stillness~~. La Conchita's streets were a mess. ~~, c~~Cars hydroplaning on the flooded roads~~, p~~. People scurrying for cover like ants from a disturbed nest. I watched as a woman, her hair plastered to her face, struggled to free her stuck umbrella,~~h~~Her eyes ~~darting~~ darted towards me with a fleeting plea for help. I looked away ~~, feeling the weight of my inaction~~. The rain-soaked world outside seemed to be shrinking, the droplets on my windshield coalescing into tiny mirrors that reflected my own uncertainty. My phone ~~, silent and unyielding,~~ lay on the passenger seat ~~, a constant reminder of my powerlessness~~. I thought of all **the what-ifs, the maybes,** the possibilities that hung in the balance. The rain intensified ~~, drumming out a rhythm that seemed to match the beat of my heart. In the chaos, I felt a strange sense of calm, as if the storm was washing away my doubts, leaving only the stark reality of the present. And yet,~~ I remained frozen, waiting for that call ~~, that spark, that would set me in motion~~. |

Table 4: An example from LAMP. Text in bold with strikethrough represents edited span. The normalized Initial and Final Writing Quality scores are 3.3 and 7.0 respectively

## A.2  Generative WRQM Prompt Formats

Table 5 shows a P and R style training prompt thats used to train WQRMs

| | |
|---|---|
| P | {"content": "You are an AI assistant who has knowledge about creative writing.", "role": "system"}

{"content": "You are given two paragraphs of writing for a given instruction.\nYour task is to determine which paragraph is overall better in terms of writing quality.\n\n Paragraph 1:\nAfter her father's passing, Marina and her family [.......]\n\nParagraph 2:\n [.......]  had cherished so deeply.\n\nYou must produce your answer in the following JSON format:\n{"preference": "1—2"}\n\n where 'preference' should be "1" if you think Paragraph 1 is better, "2" if you think Paragraph 2 is better.\n", "role": "user"}

{"content": "{"preference": "2"}", "role": "assistant"} |
| R | {"content": "You are an AI assistant who has knowledge about creative writing.", "role": "system"}

{"content": "You are given a paragraph of creative writing. You must score it on a scale from 1 to 10, where 1 is the lowest quality and 10 is the highest quality.\n\nParagraph:\nThe rich history, and recent revitalisation of Santa Barbara are most clear [.......]\n\nYou must produce your answer in the following JSON format:\n{"score": 1}\n\n where 'score' is an integer between 1 and 10.", "role": "user"}

{"content": "{"score": "7"}", "role": "assistant"} |

Table 5: Training prompts in P and R setting for instruction tuning Llama3.1 70B

## A.3   Example Rationales For Generative WQRM Training

Table 6 shows the silver natural language rationales for P and R mode.  In P its more contrastive in nature while in R its an absolute critique. Both rationales cites excerpts from the original paragarph.

| |
|---|
| Paragraph 2 is more concise and direct in its storytelling, avoiding the overwrought metaphors and clichéd language found in Paragraph 1. For example, while Paragraph 1 uses flowery phrases like "*a delicate bloom unfurling in the wake of a harsh winter*" to describe the mother's recovery, Paragraph 2 simply states "*Marina's mother slowly emerged from her all-consuming grief.*" The second paragraph also maintains emotional authenticity without becoming melodramatic. The first paragraph's phrases like "*brick by brick, memory by memory*" and "*the resilience of the human spirit*" feel forced and sentimental, whereas Paragraph 2 conveys the same emotional journey with more restraint and impact. The shorter length of Paragraph 2 also helps maintain the story's momentum without getting bogged down in unnecessary elaboration. |
| The paragraph attempts to capture a poignant moment of parent-child separation but relies too heavily on telling rather than showing, with lines like "I felt a pang of guilt only a parent could know" and "I realized I was facing my own reluctance." The emotional weight of the situation is spelled out rather than revealed through action or specific detail. While the core idea is relatable, the writing lacks distinctive imagery or memorable turns of phrase that would elevate it beyond the obvious. The final metaphor about "running up the charges to fill the space on my lighter bill" feels forced and doesn't quite land effectively. The narrative maintains a consistent tone but remains in safe, conventional territory without taking any stylistic risks that might make it more compelling. |

Table 6:  Natural language rationale for P and R modes respectively

## A.4   Datasets

**Art or Artifice** In prior work Chakrabarty et al. (2024a) evaluate writing quality in flash fiction (1,500-2,500 words). The dataset includes 12 writing prompts based on New Yorker stories, each with four responses: the original story plus three LLM-generated versions from GPT-3.5, GPT-4 and Claude v1.3. Three expert annotators ranked all four stories for each prompt, with results aggregated into majority preferences for each story pair. From the 12

prompts and all possible response pairs (4C2), the dataset contains 144 preference samples (including both AB and BA orderings). 25% are Human-AI comparisons, while 75% are AI-AI comparisons.

**LAMP-test** The LAMP corpus (Chakrabarty et al., 2024b) test set focuses on short-form creative writing (200-400 words), including fiction and non-fiction. It contains 201 triplets, each with a writing instruction and three responses: (1) AI-written, (2) AI-written+AI-edited, and (3) AI-written+AI-edited. Three professional writers ranked responses based on subjective preference, with results combined into a majority vote. For each instruction, all 3 possible response pairs were evaluated, creating 1206 total samples (by duplicating each pair in AB and BA order). Of these, 33% are AI-HumanAI comparisons, and 66% are AI-AI comparisons.

**Style Mimic** In recent work, Anonymous (2025) examined if MFA students could mimic award-winning authors' styles. Specifically, 28 MFA students were first given 20 samples written by an award-winning author (such as Haruki Murakami, Yoko Ogawa, Percival Everett, Zadie Smith, Joan Didion), along with their style verbalized in text. They were then provided with a writing instruction to recreate an original paragraph from the author (typically 200-400 words) while imitating the style of the author to the best of their ability. This data includes 150 sample pairs (student imitation vs. original author response), with the original author's work implicitly preferred. All Mirror Human samples are Human-Human comparisons. Table 7 shows an example.

**Synthetic Mirror** Prior work on AI-detection (Emi & Spero, 2024) introduced "synthetic mirrors," a two-step approach to generate writing pairs with implicit preferences. First, an LLM creates a mirror prompt from a human-written sample, extracting a plot summary and structured features (tone, style, length). Second, this prompt produces a synthetic mirror: an AI-generated response resembling the original's content and features. We selected 280 paragraphs from New Yorker flash fiction by award-winning authors (such as Alice Munro, Jhumpa Lahiri, Annie Ernaux etc). After extracting the content and structured features we devised our mirror prompts: *Write a n word paragraph in the style of author in v voice given the content below.\n plot*. We generated mirror responses using GPT-4o and Claude-3.5 Sonnet, creating 560 Human-AI pairs with implicit preference for author-written responses. The benchmark consists of 1120 total preference pairs (each duplicated in AB and BA order).

**LMArena** LM Arena Zheng et al. (2023) is an open platform for crowdsourced AI bench-marking. A recently released anonymized instructions with responses and preference judgments indicated that creative writing comprises 30% of instructions, making it one of the three most common interaction types. From 100,000 creative writing samples, we filtered for (1) English content, (2) non-tied preferences, and (3) responses between 100-2,000 words. An initial inspection of the resulting 7,981 samples revealed that many didn't match strict creative writing definitions. We further filtered noisy samples using GPT-4o, resulting in 1,959 pairs. Due to LM Arena being larger in scale than other datasets in the benchmark, we do not include both order variants (AB/BA) in the dataset but ensure that the reference order is balanced within the dataset.

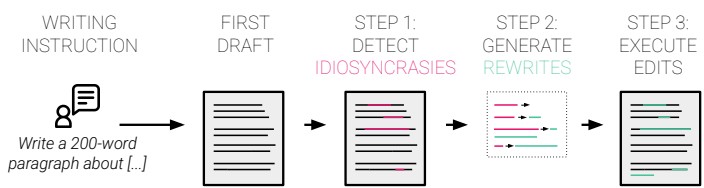

Figure 6: Three-Step Editing Pipeline to improve the writing quality of a first draft by: identifying idiosyncrasies, generating rewrites, and implementing the edits.

## A.5 Writing Quality Benchmark Difficulty Analysis

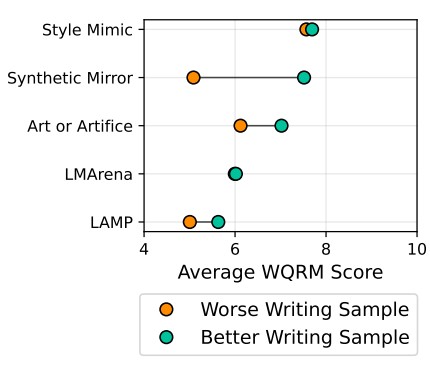

Figure 5: Gap Analysis of WQ datasets leveraging the WQRM-PR model.

In order to understand the relative difficulty of the datasets within the WQ benchmark, we performed an analysis leveraging our trained WQRM model. For each sample (consisting of two writing samples with a known human preference), we computed the WQRM score for each sample, and compiled the result for each of the five datasets in WQRM. Figure 5 plots the average of the preferred vs. less-preferred scores on each dataset.

This analysis allows to make several observations. First, the average WQRM gap is directly proportional with model performance on the benchmark. The Synthetic Mirror dataset has the largest average gap according to WQRM-PR (2.4 on average), and we find that many models achieve very close to perfect performance (98%+) on this dataset. On the other hand, the gap (according to WQRM-PR) is very small on Style Mimic (0.12) and LMArena (0.02), which aligns with many models performing at or very slightly above chance on these datasets. Second, the absolute scores for the low and high samples are indicative of the origin of the samples. Style Mimic is the only dataset to include Human-Human comparisons (both written by professionals), and the scores of both the worse and better writing samples are high (7.57 and 7.69). LMArena has a similarly small gap, but achieved with lower pair scores (5.99 and 6.02). Third, we find that the WQ dataset includes a mix of high-gap (easy) and low-gap datasets. For low-gap samples, those can be with both having lower scores (two AI-generated samples), or two high-scoring samples (two human-written samples). This confirms the breadth of evaluation included in the WQ benchmark, which is a primary objective of the WQ benchmark.

We note that this analysis should be taken with a grain of salt: the WQRM-PR model is not a perfect score predictor, and is only a proxy for analysis, since true scores would require large-scale professional annotation (which is cost-prohibitive). But this analysis matches some expectations, and provides additional evidence of the proper calibration of the WQRM-PR model, and of the breadth of evaluation in the WQ benchmark.

## A.6 Example Human Mimic Samples

Table 7 shows an Expert-MFA contrast where both paragraphs are centered around the same semantic content and writing style

## A.7 Example COT Editing Prompt

The prompt in Table 8 is generated automatically based on a sample from the LAMP dataset. An LLM is then finetuned on this prompt, effectively training it to function as a three-step editing pipeline that identifies problematic spans, rewrites the spans, and executes the edits into a final edited response.

| I watched my mother. It was March, and outside, the sun glinted off the sidewalks and the icy edges of the snow. It was Saint Patrick's Day and the nurses brought my mother a square block of green Jell-O that sat quivering on the table beside her. It was the last full day of her life, and my mother did not sleep, she did not wake. She held her eyes still and open. They were the bluest thing in the room, perhaps in all of Duluth. Bluer than the lake. They were the color of the sky on the best day of your life. My mother died fast but not all of a sudden. A slow-burning fire when flames disappear to smoke and then smoke to air. She never once closed her eyes. First they were bitter and then they were bewildered and then they changed again to something else, to a state that I have had, finally, to see as heroic. Blue, blue eyes. Daggers of blue wanting and wanting. To stay, to stay. |
| --- |
| My mother died on St. Patrick's Day. There was snow outside, and sunlight glanced off the ice and back into her hospital room. A cup of green Jell-O lingered untouched on its tray. My mother was unresponsive, but not quite asleep. Her eyes were open and vivid. There was an alertness in them still , but what she was looking at, I could not say. The bright blue of her irises was like a frozen lake, contrasting the dull wash of beige and gray shrouding the room. The sky lived inside my mother. It lived inside her then, too, even as we bided time together at the very end, knowing there was only one thing left for her to do. Her fading life was a fire flickering down to smoke—it would only last as long as there was something to burn. There was bitterness too in her eyes, then bewilderment, then a stoic, quiet heroism. A commitment to her own dignity. A promise to endure this final test, no matter how unfairly it had been imposed on her. Her eyes were so blue, my mother's eyes, a fierce blue, a frozen lake, a sheen of ice that refused to melt, even as the sun broke it apart. |

Table 7: Imitation of Original Paragraph (Top Row) from Cheryl Strayed written by an MFA student

## A.8 Expert Annotation Result Breakdown

In Table 7, we present the results of the annotations from experts for each model (GPT-4o, Llama 3.1 70b) and writing domain (fiction, nonfiction, marketing).

At a high level, the responses selected by the WQRM model (Best Edit) achieve the best average rank in all six conditions. However, the selection aligns more with expert preference (in other words, the preference is more pronounced) for the fiction domain (rather than nonfiction) and for GPT-4o responses (rather than Llama 3.1 70b). We posit that this is due to the distribution of training data for the WQRM model, which included a majority of fiction samples and did not include LLama-generated responses. However, the fact that preference is still observed on the other domains (including marketing differs widely from fiction writing) is encouraging. Improving the generalization of the WQRM further can be accomplished by collecting annotations in additional writing domains, which can be used to train an improved WQRM model.

## A.9 Comparison

Table 9 shows 3 different versions of the same paragraph. First Draft along with edited versions (Random and Best Edit) with respect rewards from WQRM. Experts rank this triplet as Best Edit > Random Edit > First Draft.

## A.10 Expert Annotation Interface

Figure 8 shows the annotation interface that is provided to experts. They read 3 responses and rank them based on overall quality.

You are given a paragraph of creative writing. Your task is to improve the quality of the writing. You must identify specific spans that can be improved, then propose rewriting for each identified span, and finally return the entire paragraph with the proposed changes implemented.

Here is the paragraph you are editing:

The room was dimly lit, with the soft hum of machinery filling the silence. I sat beside Lila, squeezing her hand, as the technician swirled the wand over her belly. The screen flickered to life, a grainy black and white, like an ancient TV trying to find it's signal. Slowly, an image began to form; the unmistakable curve of a tiny head, the flutter of something I later learned was a heartbeat. I leaned forward, almost holding my breath, mesmerized by the play of light and shadows that composed this new life. Lila's grip tightened, and I glanced at her, finding her eyes glossy, a smile tugging at the corners of her mouth. I turned back to the screen, emotions swirling 2014awe, fear, hope. In that moment, the abstract idea of öur baby ̈transformed into something real, tangible. The room's dimness seemed to fold around us, making space for this silent, wordless connection. There, in the ghostly shapes on the monitor, I saw not just a child, but a future ̌2014a future that felt both incredibly close and impossibly distant.

Part 1: Identifying Problematic Spans

Span 1: 'The room was dimly lit, with the soft hum of machinery filling the silence' (Category: 'Awkward Word Choice and Phrasing')

Span 2: 'I sat beside Lila, squeezing her hand, as the technician swirled the wand over her belly' (Category: 'Lack of Specificity and Detail')

Part 2: Proposing Rewriting for Problematic Spans

Span 1: 'The room was dimly lit, with the soft hum of machinery filling the silence' → 'The soft hum of machinery filled the room with an unfamiliar yet gentle melody.'

Span 2: 'I sat beside Lila, squeezing her hand, as the technician swirled the wand over her belly' → 'I sat beside Lila, squeezing her hand as she lay on the table while the technician swirled the wand over her rounded belly.'

Part 3: Implementing Proposed Edits

((*Written Response with Edits Executed*))

Table 8: Example COT editing instruction based on a sample from the LAMP dataset. The model is instruction to: (1) identify problematic spans, (2) rewrite the soans to improve writing, (3) execute all edits into the edited response.

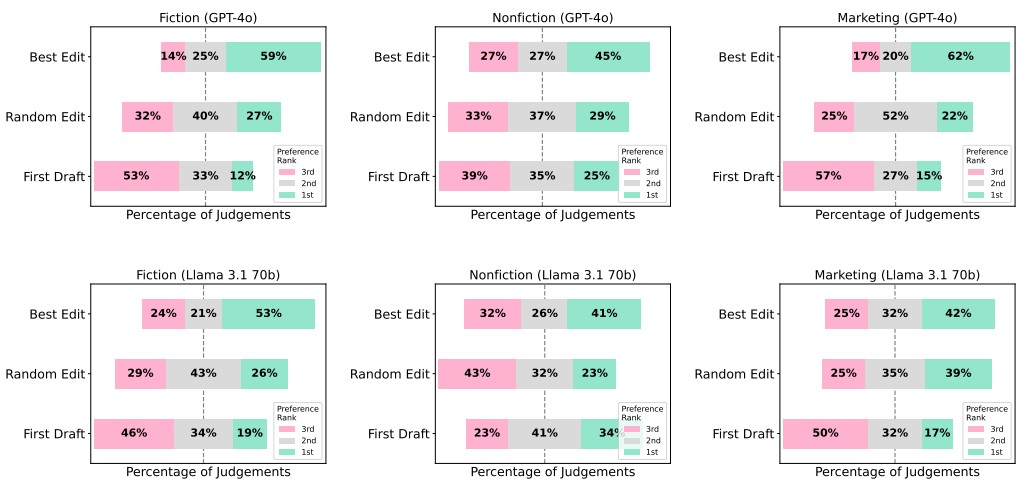

Figure 7: Breakdown of results of the expert annotation we conducted for each of the three domains (fiction, nonfiction, marketing) and the two models (GPT-4o, LLama 3.1 70b). Overall, WQRM selection was most aligned with expert preference in the Fiction domain, and for GPT-4o generations.

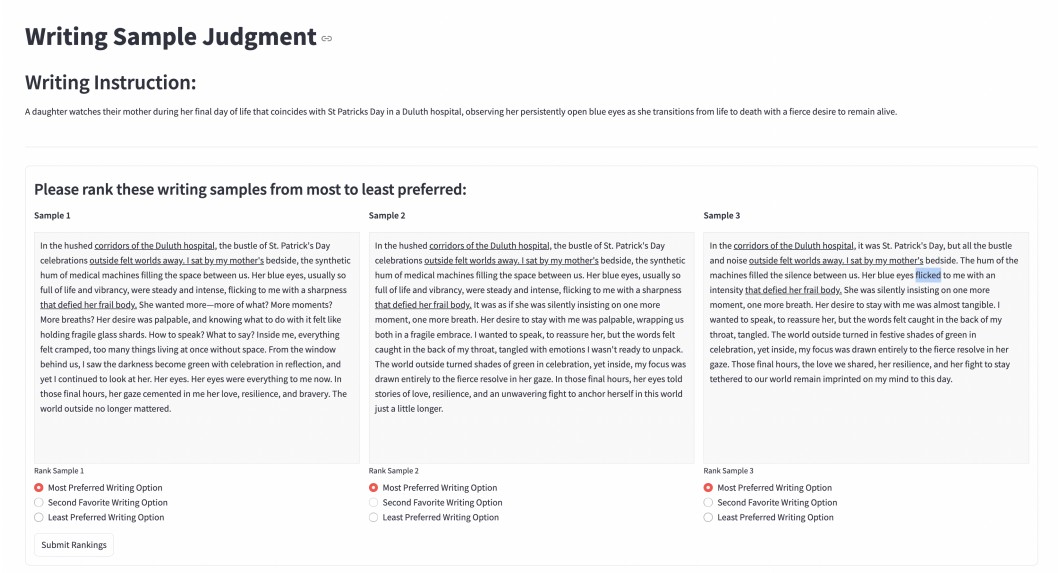

Figure 8: Annotation interface

## A.11 Better Calibrated WQRM model for Content and Quality Experiment

Since WQRM was only trained on samples from LAMP, which consists of AI-generated paragraphs edited by MFA students, it doesn't fully know how to reward higher-quality human writing. For this purpose, we added 100 paragraphs written by 5 award-winning authors (20 each) to our training data. We chose 5 authors who were part of the Style Mimic data. Each paragraph written by an award-winning author was assigned a score of 10.0. Even within writing from trained professionals, there is significant variability. To address this we source an additional 80 independent paragraphs written by MFA students published in prestigious literary magazines such as *Electric Lit, Joyland, Paris Review* and add to our training data. Each paragraph written by an MFA student was assigned a score of 7.5 [7].

---

[7]This was a design decision where 5 is average and 10 is the best, and 7.5 is a mid-point.

At 22, I handed my first completed novel to my father, not fully aware of the meaning it held for both of us. He took it, eyes softening as he turned the pages, lingering just enough to let me know he saw more than words. Then came the moment that stayed with me: an embrace that spoke volumes, a quiet prediction slipping past his lips, that one day I'd earn a place among the literary greats. Somehow, those words anchored themselves in my mind, reassuring in their sureness. Through the highs and lows of my writing career, his faith in me never wavered, echoing in my thoughts whenever doubt crept in. His death in 2002 marked an inevitable yet profound shift-suddenly, the world felt heavier. Sorting through his things, I found the suitcase he left me, an ordinary object now imbued with significance. Inside were scattered remnants of our shared past, but it was that embrace, and his words, that lingered strongest. He had seen something in me that I hadn't yet grasped myself, and for that, those moments serve as an enduring source of comfort and motivation. Years later, when I finally stood on stage accepting the literary prize he had foreseen, I wished he could have been there to witness it, though in a way he already had been all along. Dreams can be crowded with voices that spur us on or pull us back, but his was the one that guided me, quiet and steady. As I continue to write, the memory of my father's belief remains a compass, gently reminding me of where I've been and where I might yet go **"score": 3.30**

At 22, I handed my first completed novel to my father, and in that moment I became aware of the significance it held for us both. He took it and began to read, lingering just enough to let me know he saw more than words. Afterward, he embraced me and said one day I'd earn a place among the literary greats. Before then, he had said little about my writing, and these words anchored themselves in my mind, reassuring in their sureness. He had never said anything like it before, but he continued to echo that faith through the highs and lows of my career. His death in 2002 marked an inevitable yet profound shift. Suddenly the world felt heavier. Sorting through his things, I found the suitcase he left me, an ordinary object now imbued with significance. Inside were scattered remnants of our shared past, but it was that embrace and his words that lingered strongest. He had seen something in me that I hadn't yet grasped myself, and those moments served as an enduring source of comfort and motivation. Years later, when I finally stood on stage accepting the literary prize–the only prize–he had foreseen, I wished he could have been there to witness it. Dreams can be hostile to our hopes, but his was the one that guided me; his quietness was steady. Now, the memory of my father's belief remains a compass; I wish I could send him an update. **"score": 4.43**

At 22, I handed my first completed novel to my father, not fully prepared for what it might mean. He took it, eyes softening as he turned the pages, lingering long enough, I felt, to take in the feeling of things. Finally, we embraced, and he leaned back to say what I hadn't dared to hope–that one day I'd be among the literary greats. No matter how tough things got or how much death loomed over me, I was comforted by those words, almost sure of their truth. His death in 2002 brought with it an unwelcome heaviness. I found significance even in his old suitcase, which I kept, shuffling through it fondly. There were plenty of other mementos, too, but I'd always have the memory of that embrace, the words. Years later, when I finally stood on stage accepting the literary prize he'd foreseen, I wished he could have been there to witness it. Whatever noise came, whatever doubt, his voice led me quietly out of it. I swear I can still hear him now. **"score": 6.84**

Table 9: (a) First Draft (b) Random Edit (c) Best Edit along with their rewards assigned by WQRM.

Publication at a venue already means these paragraphs have undergone scrutiny and are of decent quality. After adding these 180 samples to LAMP-PR training set, we retrained WQRM.

This paragraph is written in the first person and revolves around a family Christmas gathering. The narrator reflects on how her father gave her a generous cash gift and invited her to Disney World with hiss new family. The narrator declined, fabricating an excuse about school, despite feeling the emotional distance growing between her, her father, and his new partner, Chitra. The narrators half-sisters, Rupa and Piu, were upset by this decision, not understanding why she doesn't want to join them. The narrator felt a sense of responsibility to uphold the memory of her late mother, just as Rupa and Piu symbolized their own father's legacy, while also sensing that both Chitra and her father are relieved by her decision to stay behind.The paragraph captures the emotional complexities of blended family dynamics, grief, and feelings of displacement during what should be a celebratory time.

Table 10: Detailed Content

