# OpenReview forum: "AI-Slop to AI-Polish? Aligning Language Models through Edit-Based Writing Rewards and Test-time computation"
_colmweb.org/COLM/2025/Conference — COLM 2025_

### Official Review · Reviewer_YjfD · 2025-04-22

**Rating:** 5
**Confidence:** 4
**Ethics Flag:** 1

**Summary:**

The paper introduces the Writing Quality Benchmark (WQ) and Writing Quality Reward Models (WQRM) to address challenges in evaluating and improving the quality of AI-generated text. The method leverages MBERT based models that can operate in a paragraph-tuple and individual-paragraph model. This is then adapted to generative setup using LORA to control the generated text. Experiments demonstrate strong generalization across diverse writing contexts. The paper further proposes a finetuning setup to refine a generated text at test-time.

**Questions To Authors:**

If the authors can provide details on their novel contributions and how that is different what already exists in literature, I am happy to update my rating.

**Reasons To Accept:**

The paper proposes a solid approach that is effective in its use of implicit preferences from edits for training reward models.
The outputs seem to have high accuracy and good generalization across diverse contexts.
Experimental results are strong and illustrate the promise of the approach.

**Reasons To Reject:**

While the paper proposes to introduce a benchmark, the dataset is just a collection of existing datasets - which make the benchmark less convincing a contribution.
The approaches and rewarding setups are leveraged from existing works in the literature. While the results are strong, the key novelty and contributions are unclear. Might be good if the authors can articulate more on this.

---

> ### Author Response · Authors · 2025-05-28
> **Please find our rebuttal and answers to your questions**
>
> **Benchmark Contribution** The WQ benchmark includes one newly collected dataset (Synthetic Mirror), two datasets requiring extensive processing to create pairwise preferences (Art or Artifice and LAMP-test), and one heavily filtered to focus on creative writing quality (LM Arena). Only Style Mimic was used without major modifications. This comprehensive processing work was necessary as no existing benchmark focuses specifically on writing quality assessment. Due to length limitations, these details were included in the Appendix (Appendix D1) but we will write a summary of the processing we performed in the main portion of the paper to highlight our contribution. The paper also includes expert annotations from 9 professionals (detailed in Section 6) that will be released publicly upon acceptance. Finally consolidating multiple datasets to create a benchmark has long standing evidence in NLP/AI. For example GLUE, SUPERGLUE which fueled a lot of early research on language understanding.
>
> **Modeling , Design and Experimental Contribution:** We highlight several innovative contributions:
>
> (i) using implicit preferences from edits for training reward models that perform well on benchmark datasets helps dealing with noise and inherent disagreement thats prevalent in collecting preference data for RLHF
>
> (ii) Our PR models uniquely combine scalar rewards and pairwise preferences into a single model, which empirically outperforms separate models when both signal types are available;
>
> (iii) We demonstrate small encoder-only models can compete with larger generative models;
>
> (iv) We extend rationale-based training benefits for LLM as a judge especially for subjective open-ended tasks like writing quality evaluation
>
> (v) Our test-time compute editing pipeline for over-generation and candidate selection via WRQM extends TTC benefits to tasks with non-verifiable rewards, proving its value beyond traditional math/reasoning tasks.
>
> **Practical Utility Contribution:** :  [See here](https://openreview.net/forum?id=jeDYcjuZIV&noteId=DxfyUOIMfQ)
>
> We hope that our points highlight some of the contributions of our work. We look forward to engaging further with you on the listed contributions. If our response has provided the clarity you were seeking, we would humbly request that you consider adjusting your rating accordingly.

---

> > ### Comment · Reviewer_YjfD · 2025-06-07
> >
> > Thanks for your response!

---

> > > ### Author Response · Authors · 2025-06-07
> > > **Thank you**
> > >
> > > Thank you for acknowledging. We believe we have clarified our contributions and how it is different from what already exists in literature. Since we are nearing the end of the rebuttal period if our response was adequate we humbly request that you consider adjusting your rating accordingly. Should you have further concerns or questions we are happy to engage.

---

### Official Review · Reviewer_8c7P · 2025-04-24

**Rating:** 8
**Confidence:** 4
**Ethics Flag:** 1

**Summary:**

This paper focuses on the problem of evaluating and improving the writing quality of large language model (LLM) outputs. The authors propose the Writing Quality Benchmark (WQ) from five existing writing preference datasets into a unified evaluation framework. They also introduce Writing Quality Reward Models (WQRM), trained on the LAMP dataset of AI-generated and expert-edited text pairs. These models are shown to outperform state-of-the-art LLMs and other baselines on WQ. They also develop an editing pipeline that detects writing issues, generates edits, and selects the best candidates using test-time compute (best-of-N sampling). Human evaluations with professional writers confirm that outputs ranked higher by WQRM are generally preferred, indicating the model is well-calibrated to expert judgment.

**Questions To Authors:**

* The paper builds heavily on Chakrabarty et al. (2024b), in terms of data and methodology. It would help readers if you included more details upfront—especially re: how the LAMP data was generated—rather than assuming the readers have read the paper. For example, the explanation on line 168 could be better introduced earlier in that paragraph.
* Re: MBERT-WQRM-PR: the model appears to optimize both regression and classification objectives. How were the two loss functions were combined and whether you applied any weighting scheme during fine-tuning?
* Re: the natural-language rationales: although rationales are only used during training (not available at test time), they seem to meaningfully improve performance. Some analysis on why this augmentation helps, e.g., which kinds of examples benefit most and why.
* Minor typos: Line 290: "its" → should be "it's", Line 402: "no where" → should be "nowhere"

**Reasons To Accept:**

* The paper focuses on an important and under-explored problem in real-world LLM usage: improving and evaluating writing quality.
* It introduces the Writing Quality Benchmark (WQ), a consolidated dataset that can serve as a useful resource for future research in this area.
* The proposed Writing Quality Reward Models (WQRM), trained on the LAMP dataset. They showed models fine-tuned on them achieve state-of-the-art performance on the WQ benchmark.
* The paper shows that using WQRM at test time (via best-of-N sampling) further improves writing quality, validated through expert human evaluation.

**Reasons To Reject:**

* The paper lacks a clear, formal definition of "writing quality." Without an explicit rubric or framework (e.g., fluency, clarity, coherence, style), it's difficult to interpret what the reward models are actually learning or optimizing for.
* The editing pipeline’s gains are mainly evaluated through best-of-N sampling, which improves results at the cost of increased test-time compute—potentially limiting practical applicability for latency-sensitive or low-resource settings.

---

> ### Author Response · Authors · 2025-05-28
> **Please find our rebuttal and answers to your questions**
>
> Thank you so much for the very positive evaluation of our paper.See our responses inline
>
> ## Reasons to Reject
>
> **Writing Quality Definition:** We agree with the reviewer that writing quality is challenging to define. Our definition of writing quality was based on the taxonomy introduced by Chakrabarty et al. (2024b), which itself is based on edits performed by professional writers as they improve the writing quality of text. We will specify this clearly in the updated version. A good quality written text should avoid clichés, purple prose (overly ornamental language), unnecessary exposition, poor sentence structure (very long run-on sentences), awkward words and phrases, and lack of specificity and detail. In short, it’s a combination of **fluency**, **coherence**, **clarity**, and **style**.
>
> **Efficiency:** We acknowledge that test-time computation is expensive, but it’s a newer and widely used approach for improving LLM responses. To mitigate cost, we chose a small and efficient mBERT model for the reward model, which lowers computation and improves latency to some degree. Developing _efficient_ methods that leverage WQRM to generate high-quality writing responses is an exciting avenue for future work.
>
> ---
>
> ## Answers to Reviewer
>
> 1. **Space constraints:**  We were constrained by space. In the revised draft, we will highlight the crucial details from Chakrabarty et al. (2024b) that may have been obfuscated.
>
> 2. **Joint optimization of regression & classification:**   We applied a weight of 1.0 to the classification loss and 0.25 to the regression loss, chosen so that at initialization both losses have roughly the same magnitude. We did not exhaustively explore this hyperparameter’s downstream impact and will add these details to the training description.
>
> 3. **Rationale-based improvement:**  We observed that adding natural-language rationales boosted performance most on the harder WQ subsets (Art or Artifice, Style Mimic). Table 6 in Appendix 6 provides examples of these rationales. Our design cites specific paragraph excerpts to justify judgments, making the process transparent: if a rationale cites clichés or other flaws, the model learns to penalize those traits. This aligns with recent work on “thinking before judging” [1, 2].
>
> ---
>
> ### References
>
> [1] Huang et al., *Think-J: Learning to Think for Generative LLM-as-a-Judge*, 2025.
> [2] Chen et al., *RM-R1: Reward Modeling as Reasoning*, 2025.

---

### Official Review · Reviewer_EPcP · 2025-05-13

**Rating:** 7
**Confidence:** 4
**Ethics Flag:** 1

**Summary:**

This paper introduces the methods for evaluating writing quality. Specifically, the authors trained WQRM and proposed the WQ benchmark. WQRMs are primarily categorized into two approaches: P (Preference) and R (Rating) methods, with generative RMs also considering the use of rationales. These models effectively evaluate writing quality. Furthermore, the authors demonstrate a practical application: an editing pipeline leveraging test-time computation, where WQRM selects the best among multiple edited texts, proving effective at producing higher-quality text.

**Questions To Authors:**

1. Could the authors provide information on the distribution of underlying quality score differences within the paired examples in the WQ benchmark components where such scores are available? (I understand this would require scored data.) The difficulty of the evaluation can vary depending on whether the benchmark mainly includes pairs with large or subtle quality differences.

**Reasons To Accept:**

1. This research tackles the important task of writing quality evaluation. The authors trained their WQRM model based on the LAMP dataset and validated its performance using the WQ Benchmark, which standardizes the task as a pairwise preference evaluation.

2. Section 6 is particularly interesting. The evaluation setup used to verify the quantitative score differences in WQRM (Figure 4-b) is well-designed and logically sound.

3.  Results shown in the sensitivity analysis could serve as a valuable reference for future works in writing evaluation research.

**Reasons To Reject:**

However, I have a few concerns regarding Table 2 (Writing Quality Benchmark Results) in Section 4. Since this table contains mixed results for PR, R, P,  IRO/IOR, and LLMs, the final output (the prediction values used to calculate accuracy) should be described more clearly.

1. For the R Model, authors mention that "resolving ties randomly (line 262)". However, it would be important to report how frequently ties occurred in the dataset. If the proportion of ties is high, random selection could impact the overall results. I also wonder whether the evaluation trend remains stable across different random seeds. Since this directly affects how much we can trust the results in Table 2.

2. PR Model output: Although PR models are clearly trained on both pairwise and rating data, the paper lacks clarity on how they produced their final output for evaluation. Was the direct P output used, or was preference derived from the R output? (+ It would also be interesting to know if the performance differs significantly when predicting via PR -> P versus PR -> R.)

3. When comparing the performance trends of Generative WQRM (LLaMA 3.1 70B) and Encoder-based WQRM (MBERT), there appears to be a reversal in the relationship between P and R: R < P for the generative model, but P < R for the encoder model. Given the significant performance differences, the authors should provide additional explanation for these results.

---

> ### Author Response · Authors · 2025-05-28
> **Please find our rebuttal and answers to your questions**
>
> First of all, we want to thank you for highlighting the positive aspects of our paper and asking great followup details. We are glad that you found value in our analysis in Section 6. Please find our responses inline.
>
> Addressing questions around methodological details:
>
> 1. **Regarding tie resolution for R-mode models:** We performed an analysis on the MBERT-WQRM-R model. We consider a tie if a pair's score differs by less than 0.001 (remember the scores are on a scale of 0-10). Only 4 of 4,729 pairs in the WQ test set were ties, meaning different tie-breaking methods would change results by at most 0.1%. We'll update our tie-breaking procedure description to note these cases are rare.
>
> 2. **PR -> P versus PR -> R:** Regarding the mode of use of the PR-trained models, we indeed experimented with both P-mode and R-mode, and found that R-mode use outperforms P-mode. We tested both P-mode and R-mode, finding R-mode superior. For MBERT-WQRM-PR, R-mode achieves 74.3% on WQ (reported in paper) versus 66.3% in P-mode (unreported)—an 8-point drop. We will update the discussion of the PR models to add this important nuance to the findings: models trained jointly on pairwise and reward-based datasets perform better when evaluated in R-mode in our experiments.
>
> 3. **Regarding the reversal of relationship between the P-trained and R-trained models between the MBert and Llama families:** Great observation !!  Since Llama is trained on whole number rewards only (1-10), the causal language modeling loss effectively treats this as a 10-way classification task - it's learning to predict which of the 10 possible reward tokens comes next. Meanwhile, MBert uses custom losses (pairwise classification for P, MSE for R) and can handle continuous values and is explicitly optimized for regression, allowing it to better capture the underlying reward structure even when trained on the same whole number targets. This leads the MBERT-R models to outperform MBert-P models, whereas the reverse is true for the LLama models, as they are not able to properly take advantage of the R-based data. Future work could investigate tuning Llama models with custom losses (MSE), which we hypothesize would lead to improved LLama-R models. We will add this discussion to the findings section of our results.
>
> 4. **Regarding your question about the distribution of scores for the different portions of the WQ benchmark to get a sense of level of difficulty**. This is a great idea, however as you point out only one of the five datasets in the benchmark have scored data. However, based on your suggestion we performed an additional analysis, leveraging our WQRM-PR model to "annotate" the samples with scores, and observe the distribution. See the Table below.
>
> **Table 1:** Average WQRM scores for “worse” vs. “better” writing examples across different WQ datasets.
>
> | **WQ Dataset**      | **Avg. WQRM (Worse)** | **Avg. WQRM (Better)** | **Gap** |
> |---------------------|----------------------:|-----------------------:|--------:|
> | Synthetic Mirror    |                  5.09 |                   7.52 |    2.43 |
> | Style Mimic         |                  7.57 |                   7.69 |    0.12 |
> | LAMP                |                  5.01 |                   5.63 |    0.63 |
> | Art or Artifice     |                  6.12 |                   7.02 |    0.90 |
> | LMArena             |                  5.99 |                   6.02 |    0.02 |
>
>
> **Analysis.** A few points are noteworthy:
>
> * **(4a)** The average gap is directly proportional to model performance on the benchmark. The *Synthetic Mirror* dataset has the largest gap according to *WQRM-PR* (2.4 on average), and many models reach near-perfect accuracy (98%+) on it. In contrast, the gap is very small on *Style Mimic* (0.12) and *LMArena* (0.02), which aligns with many models performing at or only slightly above chance on those splits.
>
> * **(4b)** The absolute scores for the “worse” and “better” samples reflect their origins. *Style Mimic* is the only dataset with Human–Human comparisons (both written by professionals), and both samples score highly (7.57 vs. 7.69). *LMArena* also shows a small gap, but at lower overall quality levels (5.99 vs. 6.02).
>
> * **(4c)** Together, (4a) and (4b) show that the WQ benchmark spans both high-gap (easy) and low-gap (hard) settings. Low-gap cases can be two low-scoring AI generations or two high-scoring human texts, confirming the benchmark’s breadth which is one of our key design goals.
>
> * **(4d)** We want to say *WQRM-PR* is only a proxy for true quality scores (expert annotated labels would be cost-prohibitive). Still, the trends match our expectations & validates both the calibration of *WQRM-PR* & the diversity of the WQ benchmark. We will include this analysis in the Findings section (moving the table to the Appendix for space).
>
> If our response has provided the clarity you were seeking, we would humbly request that you consider adjusting your rating accordingly.

---

> > ### Comment · Reviewer_EPcP · 2025-06-05
> >
> > Thanks for your comments. I found responses 3 and 4 particularly interesting. It would be great to train MSE with LLaMA and see how it compares! Also, your response to point 4 addressed my remaining concerns.
> >
> > If all the points you mentioned in your response were included in the paper, I think the contribution would be much clearer and easier to follow. I’ll update my score.

---

> > > ### Author Response · Authors · 2025-06-05
> > > **Thank you**
> > >
> > > Thank you for the acknowledgement and increasing our scores. We will make sure to mention all these details in the camera ready.

---

### Official Review · Reviewer_Ac9o · 2025-05-13

**Rating:** 7
**Confidence:** 3
**Ethics Flag:** 1

**Summary:**

The paper introduces the Writing Quality Benchmark, a combined test built from five preference datasets spanning AI-AI, AI-human, and human-human comparisons. Current large language models perform only slightly above chance on this benchmark. The authors train a Writing Quality Reward Model on expert edits from the LAMP corpus and reach 74 % accuracy. They plug this reward into a three-step editing pipeline with best-of-20 sampling. In a study with 9 professional writers over 100 prompts, the reward-ranked edit is preferred 66 % of the time, rising to 72 % when the reward gap is large.

**Questions To Authors:**

Please consider citing the following paper as it is very relevant to your work:

arXivEdits: Understanding the Human Revision Process in Scientific Writing, EMNLP 2022

**Reasons To Accept:**

The work focuses on writing quality, a dimension where language models still lag, and delivers both a public benchmark and a task-specific reward model. The empirical design is clear: data sources, training procedure, and evaluation metrics are specified in enough detail for replication. The human study, though modest in size, shows that the reward aligns with expert judgement.

**Reasons To Reject:**

The evidence for generality is limited. The human study involves just 9 professional writers and 100 prompts, providing only a narrow sample of genres and styles. Because both training and evaluation draw heavily on LAMP edits, the reward model may have learned domain-specific cues that do not transfer to other writing contexts. The paper also compares its system only to standard LLM judges, omitting commercial editing tools and recent self-refinement methods, so the practical advantage remains uncertain.

---

> ### Author Response · Authors · 2025-05-28
> **Please find our rebuttal**
>
> First of all, we want to thank you for highlighting the positive aspects of our paper. Please find our responses inline.
>
> **Evidence for Generality:**   The WQ benchmark was constructed specifically with generalization in mind. Our benchmark consists of AI-AI, Human-AI, Human-Human and AI-HumanAI pairs, covering all possible combinations for quality judgment. Our models were trained on AI and HumanAI paragraphs but can still transfer to purely AI-written or purely human-written paragraphs, showing signs of generalization. Not only that, our models can also do writing-quality judgments for long form / long context (something it wasn’t trained on). The paragraph length for _Art or Artifice_ is 5–6× longer than the average paragraph length in training. See also _(Some Additional Experiments highlighting Practical Utility of WQRM)_ for evidence of generalization.
>
> **Human Evaluation sample size :**   We conducted the reward-model calibration study with 9 experts (which adds credibility to our approach). For tasks such as writing, it’s often better to have a smaller group of highly qualified expert annotators compared to a larger pool of non-experts. This has also been the trend in several human-centered studies focusing on writing [^1][^2].
>
> **Domain Coverage:**  Our test data for WQ consists of literary fiction, creative nonfiction (cooking, travel, internet advice) and our additional human evaluation (100 prompts: literary fiction, news writing, marketing). We think this constitutes a broad representation of domains and genres. Unfortunately, covering every writing task would be cost-prohibitive, so we selected domains that are common and popular. Thank you for sharing arXivEdits; it’s indeed relevant and we will absolutely add a citation. Although we considered scientific writing, early rewriting experiments often hallucinated or produced inaccuracies, so we deprioritized it. We’ll note this in our Limitations.
>
> **Comparison with Commercial Tools / Self Refinement:**   We’re not aware of any commercial tool that judges writing quality. Our focus is on learning robust rewards for quality rather than self-refinement. Indeed, Section 5.1 uses targeted self-refinement but without knowing what to refine (or if intermediate refinements truly improve quality), you risk reward hacking [^3].
>
> **Practical Utility Contribution:** See [Here](https://openreview.net/forum?id=jeDYcjuZIV&noteId=DxfyUOIMfQ)
>
> ---
>
> [^1]: People who frequently use ChatGPT for writing tasks are accurate and robust detectors of AI-generated text. _ACL 2025_ (5 experts)
> [^2]: _Art or Artifice? Large Language Models and the False Promise of Creativity_. _CHI 2024_ (10 experts)
> [^3]: Spontaneous Reward Hacking in Iterative Self-Refinement — Jane Pan, He He, Samuel R. Bowman, Shi Feng

---

> > ### Comment · Reviewer_Ac9o · 2025-06-05
> >
> > Thank you for your response!

---

> ### Author Response · Authors · 2025-06-05
> **Thank you**
>
> Thank you for acknowledging our response. If we have addressed your concerns, we would humbly request that you consider adjusting your rating accordingly. We are happy to engage should have any further comments.

---

### Author Response · Authors · 2025-05-28
**Some Additional Experiments highlighting Practical Utility of WQRM**

Effectively judging writing quality has a broader impact on both understanding and improving LLM writing. Writing quality, however, is often closely tied to content.

Using Style Mimic's 50 writing instructions, we created two variants: brief prompts (30 words) with minimal content detail, and detailed prompts (150-200 words) with rich content. Each prompt included both an original excerpt from an award-winning author and an MFA student's attempt to mimic that style.**With regards to novelty and practical utility of WQRM models we did an experiment that better understands the impact of content for writing quality estimation**

WQRM, trained only on AI-generated paragraphs edited by MFA students, may not be sufficient to evaluate high-quality human writing. We enhanced the training data with 100 paragraphs from 5 award-winning authors (scored 10.0) and 80 paragraphs from MFA students published in prestigious magazines like Electric Lit and Paris Review (scored 7.5). Publication at a venue already means these paragraphs have undergone scrutiny and are of decent quality. After adding these 180 samples to the LAMP-PR dataset, we retrained WQRM to better assess writing quality across different skill levels.

| Model/Author           | Detailed Prompt Mean Scores | Brief Prompt Mean Scores |
| :-                      | :-:                        | :-:                       |
| Award-winning Author   | $8.82 \pm 0.247$           | $8.82 \pm 0.247$          |
| MFA Student            | $8.52 \pm 0.355$           | N/A                       |
| GPT-4.5-preview        | $6.86 \pm 0.749$           | $5.52 \pm 0.877$          |
| GPT-4o                 | $6.07 \pm 1.111$           | $5.06 \pm 0.827$          |
| GPT-4o-mini            | $5.61 \pm 1.236$           | $4.49 \pm 0.805$          |
| GPT-3.5-turbo          | $4.93 \pm 0.898$           | $4.50 \pm 0.774$          |
| Claude 4 Opus          | $7.42 \pm 0.647$           | $5.96 \pm 0.960$          |
| Claude 4 Sonnet        | $7.14 \pm 0.723$           | $5.73 \pm 0.846$          |
| Claude 3.7 Sonnet      | $7.34 \pm 0.607$           | $5.93 \pm 0.787$          |
| Claude 3.5 Sonnet      | $6.70 \pm 0.832$           | $5.67 \pm 0.822$          |
| Claude 3 Haiku         | $5.94 \pm 1.173$           | $4.43 \pm 0.804$          |
| Gemini 2.5 Pro         | $7.06 \pm 0.941$           | $5.34 \pm 0.942$          |
| Gemini 2.0 Flash       | $6.94 \pm 1.101$           | $4.90 \pm 0.903$          |
| Gemini 1.5 Pro         | $7.02 \pm 1.033$           | $5.22 \pm 0.852$          |
| Gemini 1.5 Flash       | $6.33 \pm 1.087$           | $4.95 \pm 0.942$          |


Since WQRM was only trained on AI-generated paragraphs edited by MFA students and data from only 5 award-winning authors, the ability of the model to score 50 author-written texts higher than all LLMs (some even very new ) **provides evidence of generalization and practical utility of the model**. This also shows that we can augment WQRM with newly annotated data from other domains/genres for better assessment in future

---

### Decision · Program_Chairs · 2025-07-08

**Decision:**

Accept

**Comment:**

The paper concerns evaluating and improving the quality of AI-generated writing. The authors introduce a benchmark (comprising new datasets and existing reprocessed datasets) for writing quality judgments. They evaluate LLMs as writing quality judges, identifying weaknesses in otherwise strong models in assessing writing quality. The paper then introduces supervised writing quality reward models that can be used to select high quality drafts from multiple candidates in an editing pipeline. Human evaluation with a small group of experienced writers shows moderate agreement with the supervised reward models.

The work tackles an important task where LLMs lag human writers, which is currently underexplored. The public benchmark is valuable and contains newly produced data, as compiling existing datasets (with modifications) across diverse genres, styles and combinations of AI and human-written text. There is also an interesting contribution in the design of task-specific reward models, which are shown to work effectively (SotA; smaller encoder models rather than LLMs), align reasonably well with expert judgment, and improve quality effectively as part of an editing pipeline . The work is presented clearly with sufficient details of experiments, which provide several useful insights. The idea of using implicit preferences from edits is itself novel, as far as I can tell.

There is some information from the rebuttals that should be included (or clarified) in the final paper if accepted:
-	A clearer definition of writing quality
-	The explanation for the performance trends of generative and encoder-based WQRM models
-	The table of WQRM scores for worse and better examples, and the accompanying analysis.
-	The results for author-written texts showing further generalisation of the model.

Considering the paper’s limitations:
- The number of human evaluators is small (9), but as these are expert annotators, the results are still meaningful, even though they cannot represent the full range of opinions on writing quality.
- The reward model may not generalize to other writing contexts and could be fitted to domain-specific cues – but the LAMP dataset is designed to be very broad across many genres, so, while broader evaluation could strengthen the results, I don’t believe this is a major concern.
- Self-refinement methods are not tested against this approach, but the focus of this paper is somewhat complementary.